# Bioadhesive Eudragit RL^®^100 Nanocapsules for Melanoma Therapy: A Repurposing Strategy for Propranolol

**DOI:** 10.3390/pharmaceutics17060718

**Published:** 2025-05-29

**Authors:** Naomi Gerzvolf Mieres, Soraia de Oliveira Simião, Luiza Stolz Cruz, Rafaela Cirillo de Melo, Najeh Maissar Khalil, Juliana Sartori Bonini, Fabiane Gomes de Moraes Rego, Marcel Henrique Marcondes Sari, Roberto Pontarolo, Raul Edison Luna Lazo, Jéssica Brandão Reolon, Luana Mota Ferreira

**Affiliations:** 1Programa de Pós-Graduação em Ciências Farmacêuticas, Universidade Federal do Paraná, Curitiba 80210-170, Brazil; nmgerzvolf@gmail.com (N.G.M.); soraiasimiao@ufpr.br (S.d.O.S.); rafaelacirillo@ufpr.br (R.C.d.M.); rego@ufpr.br (F.G.d.M.R.); marcelsari@ufpr.br (M.H.M.S.); pontarolo@ufpr.br (R.P.); raulunalazo@gmail.com (R.E.L.L.); 2Departamento de Farmácia, Universidade Estadual do Centro-Oeste, Guarapuava 85040-167, Brazil; luizastolz@gmail.com (L.S.C.); juliana.bonini@gmail.com (J.S.B.); 3Applied Nanostructured Systems Laboratory, Universidade Estadual do Centro-Oeste, Guarapuava 85040-167, Brazil; najeh@unicentro.br

**Keywords:** drug repurposing, polymeric nanoparticles, topical drug delivery, cutaneous cancer therapy

## Abstract

**Background/Objectives:** Cutaneous melanoma is a potent neoplasm whose advancement is linked to catecholamine-induced angiogenesis through β-adrenergic receptors. Propranolol (PROP), a non-selective β-blocker, holds potential in oncology, but its systemic side effects restrict its viability. This study aims to nanoencapsulate PROP in Eudragit RL^®^100 polymeric nanocapsules for topical melanoma treatment. **Methods**: Nanocapsules were created through interfacial deposition of preformed polymer and characterized in terms of particle size, zeta potential, pH, drug content, and encapsulation efficiency. In vitro evaluations include release profile, antioxidant activity, bioadhesiveness, hemolysis, cytotoxicity, and antitumor effect on melanoma cells. Additionally, migration assays were conducted. **Results**: The nanocapsules displayed an acidic pH, an average size of 151 nm, and a positive zeta potential. An encapsulation efficiency of 81% was achieved, even with the hydrochloride form of the drug. The release profile exhibited sustained release of PROP, showcasing enhanced antioxidant activity in the nanoencapsulated form. The formulations also exhibited significant bioadhesion with mucin and an in vitro hemolysis rate over 50%, attributed to the cationic polymer and surfactants present. Moreover, in the cell viability assays, the NC-PROP formulations significantly reduced melanoma cell viability. In the migration assay, both the nanocapsules with and without the drug significantly inhibited cell migration, supporting the potential therapeutic benefits of these formulations. **Conclusions**: The nanoencapsulation of PROP in Eudragit RL^®^100 presents a viable strategy for topical treatment of cutaneous melanoma, enhancing release duration and reducing systemic effects. The assessments indicated distinct physical properties and substantial therapeutic potential.

## 1. Introduction

Melanoma is considered one of the most aggressive types of skin cancer, originating from the uncontrolled proliferation of melanocytes and, if not detected early, can metastasize, drastically worsening the prognosis [1]. Metastatic melanoma poses a major clinical challenge, with nearly 50% of patients developing liver metastases [2,3]. This progression severely diminishes survival rates, dropping five-year survival to 15–20% [1,2]. Given its high metastatic potential, therapies that halt or slow tumor progression are urgently needed.

Evidence suggests a strong connection between stress-related catecholamines and tumor growth progression, mediated by β-adrenergic receptors (β-ARs), particularly the β2 subtype [4,5,6]. β-ARs are membrane receptors activated by catecholamines such as epinephrine and norepinephrine, which trigger signaling pathways associated with angiogenesis, including the nitric oxide synthase and the release of proangiogenic factors such as VEGF, PDGF, and SDF-1, contributing to melanoma progression. In solid tumors, pathological angiogenesis sustains the malignant cells nourished and vasculated [7,8,9].

Drug repurposing has emerged as a cost-effective strategy to identify new applications for approved drugs with good safety profiles, making them feasible for rapid implementation in new medical applications [10]. Propranolol (PROP), a non-selective β-blocker initially used for cardiovascular diseases, has been explored in other contexts such as essential tremor, migraines, infantile hemangiomas, wound healing in burn patients, and certain cancers, including melanoma and angiosarcoma [11,12,13,14]. Although its anti-angiogenic mechanism is still unclear, PROP appears to reduce endothelial cell proliferation and migration, inhibit vasodilation, and suppress VEGF and basic fibroblast growth factor (bFGF) expression, contributing to reduced tumor vascularization and growth [15].

Although PROP demonstrates promising pharmacological effects, its systemic use carries risks because of its ability to cross the blood–brain barrier, potentially leading to neurocognitive impacts. Furthermore, this drug can induce respiratory, cardiac and metabolic disturbances, disrupt sleep, alter behavior, and impair memory [13,16,17,18,19]. From a pharmaceutical perspective, propranolol presents multiple formulation challenges. Its high first-pass metabolism leads to low oral bioavailability, and its relatively short half-life demands frequent administration to maintain therapeutic levels [20,21,22,23]. Moreover, its hydrophilic hydrochloride salt form exhibits limited permeation through the stratum corneum when applied topically, making skin delivery particularly difficult [24]. In addition, propranolol is chemically unstable under light exposure due to its photosensitive nature, which can compromise product shelf-life and efficacy [25]. These limitations necessitate the development of innovative delivery systems that can enhance skin penetration, protect the drug from degradation, and promote sustained local release while minimizing systemic exposure.

Topical drug delivery systems offer an alternative that could overcome many of these limitations, particularly for the treatment of cutaneous melanoma. Among the advanced delivery platforms, polymeric nanocapsules (NCs) have shown great potential due to their ability to protect labile drugs, prolong release, and enhance skin penetration [26]. Nanocapsules comprise an oily core surrounded by a polymeric shell, where the drug is usually dispersed, dissolved in, or even adsorbed onto the surface [17]. Eudragit RL^®^100 is a biocompatible, bioadhesive, and water-insoluble polymer that offers sustained release and improved permeation profiles [27,28]. Encapsulating PROP in Eudragit RL^®^100 nanocapsules could enhance its physicochemical stability, prevent degradation, extend half-life, and facilitate a controlled release directly at the tumor site.

Furthermore, the cationic character of Eudragit RL^®^100 enables stronger interaction with the skin surface, which is predominantly anionic under physiological conditions. This electrostatic interaction favors the bioadhesion of the nanocapsules to the stratum corneum, leading to prolonged residence time and enhanced localization of the drug at the application site [29,30]. These bioadhesive formulations are especially advantageous in topical melanoma therapy because they can increase drug retention near the basal layer of the epidermis, where melanocytes reside and malignant transformation initiates. By promoting intimate and prolonged contact with the viable epidermis, bioadhesive nanocapsules may enhance local drug bioavailability, maximize therapeutic efficacy against melanoma cells, and minimize systemic absorption and associated side effects [29,31,32].

Currently, PROP is only available in oral and intravenous forms, which limits its application in localized therapies and increases the likelihood of off-target effects. While topical PROP formulations have been researched, they primarily target infantile hemangiomas and are often hydrophilic, restricting local drug effects [33,34]. This research pioneers the use of Eudragit^®^ nanocapsules for PROP, enabling controlled release directly at the tumor site. To the best of our knowledge, only one previous study has explored Eudragit RS^®^-based nanoformulations of PROP; however, that study focused on nanobeads rather than nanocapsules, further underscoring the novelty and innovation of our approach [35].

In this context, the present study proposes developing and characterizing propranolol-loaded Eudragit RL^®^100 nanocapsules for topical application as an innovative therapeutic strategy for cutaneous melanoma. The central objective is to repurpose propranolol by overcoming its pharmacokinetic and physicochemical limitations—such as its short half-life, poor skin permeability, and systemic side effects—through encapsulation in a polymeric nanocarrier. By leveraging the bioadhesive and controlled release properties of Eudragit RL^®^100 nanocapsules, this formulation aims to enhance local drug retention at the tumor site, prolong drug action, and minimize systemic absorption. Ultimately, this approach seeks to improve the efficacy and safety profile of propranolol for the localized treatment of skin cancer.

## 2. Materials and Methods

### 2.1. Material

Propranolol hydrochloride (SM Empreendimentos, São Paulo, Brazil) was donated by Vicofarma (Guarapuava-PR, Brazil). Eudragit RL^®^100 was kindly donated by Evonik (São Paulo, Brazil). Span^®^ 80 (sorbitan monooleate) and Tween^®^ 80 (polysorbate 80) were purchased from Sigma Aldrich (São Paulo, Brazil). Medium-chain triglycerides (MCT) mixture was supplied by Delaware (Porto Alegre, Brazil). HPLC-grade methanol was acquired from Tedia (Rio de Janeiro, Brazil). All other solvents and reagents were analytical grade and used as received.

### 2.2. Analytical Method

The PROP quantification was carried out using a high-performance liquid chromatography (HPLC) method (Agilent 1100 series, Santa Clara, CA, USA) equipped with a pump (QuatPump G1311A), a degasser (G1379A), an automated sampling, an ALSTherm G1330B, a ColCom G1329A, and a diode array detector (DAD—G1315B). The chromatographic conditions were employed according to the Brazilian Farmacopeia, 7th ed. The separation was performed in a column RP C8 (150 mm × 4.60 mm, 5 μm), using a mobile phase composed of 0.15 M phosphoric acid and sodium lauryl sulfate in water, acetonitrile (ACN), and methanol (28:36:36), at a flow rate of 1.5 mL/min and injection volume of 20 μL. All samples and the mobile phase were filtered through a 0.45 µm filter. The detection was evaluated at a wavelength of 290 nm [36]. The method was linear in the concentration range of 5 to 50 µg/mL, with a limit of detection and quantification of 1.13 and 3.42 µg/mL, respectively.

### 2.3. Nanocapsule Preparation

PROP-loaded nanocapsules (PROP-NC) were prepared using the interfacial deposition of the preformed polymer method [28,37]. For this, an organic phase containing Span 80^®^ (0.077 g), Eudragit RL^®^100 (0.1 g), MCT (330 μL), PROP (0.01 g), and acetone (27 mL) was kept under moderate magnetic stirring at 40 °C. After 60 min, this organic phase was injected into 53 mL of a Tween^®^ 80 aqueous dispersion (0.077 g) and kept under magnetic stirring at room temperature. After 10 min, the acetone and part of the water were eliminated by evaporation under reduced pressure to achieve a 10 mL final volume, corresponding to 1 mg/mL of PROP. For comparison purposes, formulations without PROP were also prepared (NC-B).

### 2.4. Physicochemical Characterization

#### 2.4.1. pH Determination

The pH values of the nanocapsule dispersions were determined directly from the formulations using a previously calibrated potentiometer (SevenEasy^®^, Mettler Toledo, Singapore) at room temperature (25 ± 1 °C).

#### 2.4.2. Mean Particle Size, Polydispersity Index, and Zeta Potential

The mean particle sizes and polydispersity indexes (PDI) were measured by photon correlation spectroscopy after dilution of an aliquot of dispersions in purified water (1:500) (Zetasizer Nanoseries, Malvern Instruments, Cambridge, UK). After diluting the samples with purified water, the surface charge was evaluated by zeta potential analyses performed with the same equipment (1:500).

#### 2.4.3. Drug Content and Encapsulation Efficiency (EE)

PROP content was determined by the HPLC method as previously described (Section 2.2). To prepare the samples, an aliquot of the formulation was diluted in 1 mL of a 1:1 (*v*/*v*) acetonitrile/acetone mixture to dissolve the polymer, then brought to a final volume of 10 mL with mobile phase. The solution was sonicated for 10 min to ensure complete drug extraction. Then, the samples were filtered through a 0.45 μm regenerated cellulose membrane and injected into the HPLC system. To determine encapsulation efficiency (EE%), 300 µL of nanocapsules were submitted to ultrafiltration/centrifugation using centrifugal devices (Amicon^®^ 10.000 MW, Millipore, Burlington, MA, USA) at 2200× *g* for 10 min. Free propranolol was determined in the ultrafiltrate. The EE% was calculated from the difference between the total and free drug concentrations using the described HPLC method.

### 2.5. In Vitro Drug Release

The dialysis diffusion technique was used to determine the PROP release profile from NCs. An aliquot of 4 mL of NC-PROP, corresponding to 4 mg, was placed in a dialysis tubing cellulose membrane (MWCO 10,000, SpectraPor 7, Sigma Aldrich, São Paulo, Brazil). The membrane was immersed in 600 mL of release medium, consisting of acetate buffer pH 5.5, maintained at 37.0 ± 1 °C under continuous stirring at 50 rpm. At predetermined time intervals (5 min, 15 min, 30 min, 45 min, 1 h, 2 h, 3 h, 4 h, 8 h, and 12 h), aliquots of the dissolution medium were collected and analyzed by HPLC to determine the percentage of drug released. The release medium was replaced to maintain sink conditions. For comparison purposes, a PROP solution (PROP, 1 mg/mL), prepared in water, was also evaluated under the same conditions.

### 2.6. Antioxidant Activity

The antioxidant effect of the NC-PROP was evaluated through the ability to scavenge the synthetic radical 2,2′-azinobis (3-ethylbenzothiazoline-6-sulfonic acid) (ABTS+), as previously described by Re, R (1999) [38], with some modifications. The ABTS⁺ radical solution was prepared by mixing ABTS stock solution (7 mM) with sodium persulfate (140 mM), 12 h before the test. A PROP in aqueous solution (PROP, 1 mg/mL), NC-PROP, and NC-B were diluted in distilled water to reach concentrations of 0.5, 1, 5, 25, 50, and 100 μg/mL and incubated at room temperature with the ABTS solution for 30 min under light protection. An ABTS+ solution was kept under the same reaction conditions and used as a negative control; an ascorbic acid solution was used for the positive control. Absorbance was measured in a spectrophotometer at 734 nm. Additionally, the different treatments, in their varying concentrations, were diluted in water instead of the ABTS⁺ solution, and the absorbance of these samples was also measured, serving as a “blank” to consider the interference of the opalescence of the nanocapsules during readings. Radical scavenging activity was expressed as a percentage of the elimination capacity compared to the negative control, according to Equation (1).Scavenging Capacity (%) = 100 − [(Abs A − Abs B)/Abs C) × 100],(1)
where Abs A is the absorbance of the sample incubated with ABTS+, Abs B is the absorbance of the blank, and Abs C is the absorbance of the negative control.

### 2.7. Bioadhesiveness

The adhesiveness of the nanocapsules was evaluated using the mucin particle method [39]. A dispersion made of porcine type II mucin in ultrapure water (0.1% *w*/*v*) was evaluated for size analysis and zeta potential. Nanocapsule dispersions were diluted in mucin dispersion (1:500), and the size and zeta potential were re-analyzed in ZetaSizer (Malvern Instruments, Malvern, UK), as described previously (Section 2.4).

### 2.8. Hemolysis Assay

A direct contact assay was performed following the guidelines outlined in the Standard Practice for Assessment of Hemolytic Properties of Materials [40], utilizing human erythrocytes to assess the biocompatibility of the nanoformulations. The experimental protocol received prior approval from the Research Ethics Committee of the Federal University of Paraná, Brazil (approval number #43948621.7.0000.0102). Peripheral blood samples were collected from healthy human volunteers in heparinized tubes and centrifuged at 2000 rpm for 5 min using a Centribio 80-2B centrifuge (Biosystem, Campinas, Brazil). The plasma layer was removed, and the remaining red blood cells were washed three times with a 0.9% NaCl solution. A 10% hematocrit dispersion was then prepared with a 0.9% NaCl solution.

Aliquots of the samples NC-PROP, NC-B, and PROP in aqueous solution (PROP, 1 mg/mL) were diluted to concentrations of 0.1, 0.2, 0.3, 0.5, 0.8, and 1.0 µg/mL and added to microtubes containing 800 µL of the 0.9% NaCl solution. Following this, 100 µL of the redispersed erythrocyte solution was introduced into each tube. Positive and negative controls were established using ultrapure water and 0.9% NaCl solution, respectively. Blank samples containing the colloidal dispersion without erythrocytes were also prepared in microtubes containing 900 µL of the 0.9% NaCl solution. All samples were incubated at 37 °C for 1 h and then centrifuged at 1200 rpm for 5 min, after which the absorbance of the supernatant was measured at 540 nm using a UV-1800 spectrophotometer (Shimadzu, Kyoto, Japan). The hemolysis percentage was calculated according to Equation (2).Hemolysis (%) = [(Abs A − Abs B)/Abs C) × 100],(2)
where Abs A: sample absorbance; Abs B: blank absorbance; and Abs C: positive control absorbance.

### 2.9. Cell Viability Assay

L-929 fibroblasts and B16-F10 melanoma cells were acquired from Rio de Janeiro Bank Cell, Codes 0188 and 0046, respectively. They were maintained under standard culture conditions (37 °C, 5% CO_2_) in a humidified incubator (Model MCO-170ACL-PA, PHC Corporation, Moriguchi, Japan). The cells were grown in Dulbecco’s Modified Eagle Medium (DMEM), supplemented with 10% fetal bovine serum (FBS) and 1% antibiotic solution containing penicillin (10,000 U/mL) and streptomycin (10 mg/mL).

For the cytotoxicity assay, cells were seeded at a density of 1 × 10^4^ cells per well (equivalent to 1 × 10^5^ cells/mL) in 96-well plates and allowed to adhere overnight. Treatments included NC-B, NC-PROP, and PROP, each diluted in medium to achieve final PROP-equivalent concentrations of 0.5, 1.0, 5.0, 25, and 50 μg/mL. Untreated cells cultured in medium served as the negative control. After 24 h of exposure, the DMEM was removed and replaced with 100 μL of MTT solution (0.5 mg/mL in DMEM). Plates were incubated for an additional 2 h under the same culture conditions. Subsequently, the medium was discarded, and formazan crystals were dissolved in 100 μL of dimethyl sulfoxide (DMSO). Absorbance was measured at 570 nm using a Cytation 5 microplate reader (BioTek Instruments, Winooski, VT, USA). Cell viability was expressed as a percentage relative to the negative control. Each treatment was tested in triplicate, in at least three independent experiments.

### 2.10. Scratch Wound Assay

The evaluation of cell proliferation was performed using a scratch wound assay adapted from the protocol described initially by Bürk (1973) [41]. B16-F10 cells were seeded at a density of 3 × 10^5^ cells/well in 96-well plates, using DMEM supplemented with 10% FBS, and incubated under standard conditions (37 °C, 5% CO_2_) for 24 h to allow the formation of a confluent monolayer. Following confluency, a linear wound was created across the center of each well using a sterile pipette tip (10 µL). The dislodged cells were gently removed by aspiration, and the wells were rinsed with sterile phosphate-buffered saline (PBS). The cells were treated with the test samples (NC-B, NC-PROP, and PROP) at 0.5, 1, and 5 μg/mL concentrations. DMEM served as the negative control.

Images were captured using phase contrast microscopy at 10× magnification (Cytation 5, BioTek Instruments, Winooski, VT, USA) immediately after treatment application (T_0_) and subsequently at 6 h intervals over 24 h (T_6_, T_12_, and T_24_). Cell proliferation was quantified by area measurement. The area reduction was compared over time and between treatments, and the results were expressed as a percentage relative to T_0_.

### 2.11. Statistical Analysis

Formulations were prepared and analyzed in triplicate (*n* = 3). The results are presented as the mean ± standard deviation (SD) or standard error of the mean (SEM). The normality of data distribution was assessed using the D’Agostino normality test. Subsequently, a *t*-test or one- or two-way analysis of variance (ANOVA), was conducted to evaluate the statistical significance, followed by the Newman−Keuls test based on the specific experimental design. Significance was defined as *p* < 0.05. GraphPad Prism^®^ version 8 statistical software was employed for all statistical analyses and figure generation.

## 3. Results and Discussions

The treatment of melanoma remains challenging, as conventional therapies such as surgery, radiation, and chemotherapy often fail to deliver sufficient drug concentrations to tumor sites while inducing severe systemic toxicity [42,43]. The non-specific cytotoxicity of chemotherapy damages both cancerous and healthy cells, frequently leading to adverse effects that compromise treatment adherence. These limitations underscore the need for targeted drug delivery systems to minimize off-target toxicity and improve therapeutic outcomes [43,44].

Given these challenges, alternative approaches for melanoma treatment have been explored, including the repurposing of established drugs with favorable safety profiles, such as β-blockers and antidiabetic agents [10]. A retrospective study revealed that patients with cutaneous melanoma undergoing β-blocker therapy exhibited reduced disease progression and lower mortality rates compared to non-exposed individuals [14].

Propranolol, in particular, has demonstrated anti-angiogenic effects in prior studies [45]. Topical PROP formulations have emerged as a promising strategy to enhance drug delivery, supporting their potential use in vascular tumors and hemangiomas, although clinical validation remains pending [46,47]. Combining localized topical therapy with nanostructured systems could enhance treatment adherence by mitigating systemic side effects and reducing dosing frequency through prolonged drug release [48,49]. Such an approach would maximize drug accumulation at the tumor site while minimizing systemic exposure, improving efficacy and patient compliance.

### 3.1. Physicochemical Characterization

The NC-PROP dispersions appeared opalescent and white, with no visible precipitates. As shown in Table 1, the formulations had an acidic pH (around 3.85 ± 0.11), a mean particle size of 151 ± 7.86 nm, a low PDI of 0.134 ± 0.01, and a positive zeta potential (25.64 ± 4.9 mV). In addition, NC-B exhibited similar physicochemical properties, with a particle size of 150 ± 1.74 nm and a slightly higher PDI (0.166 ± 0.03), indicating a monodisperse system in both cases. However, NC-B showed a higher zeta potential (34.13 ± 8.70 mV), which may be associated with the absence of drug interaction with the polymeric wall [26,50]; however, no statistical difference was observed between the values (*p* > 0.05). Zeta potential evaluation confirmed the cationic nature of both nanocapsules, attributed to the Eudragit RL^®^100, a positively charged polymer, suggesting good colloidal stability [27]. A high absolute zeta potential, whether positive or negative, contributes to particle repulsion and prevents aggregation [32].

The small particle sizes obtained favor topical drug delivery, as they enhance skin permeation by increasing surface area. The presence of Tween^®^ 80 likely contributed to the reduced particle size through steric stabilization, consistent with its surfactant properties [27,43,51]. The close similarity between NC-PROP and NC-B suggests that the incorporation of propranolol did not significantly alter the structural characteristics of the nanocapsules (*p* > 0.05).

The particle size of approximately 150 nm is particularly significant for topical treatments aimed at cutaneous melanoma. Nanocarriers smaller than 200 nm can penetrate the stratum corneum, reaching deeper layers of the epidermis and even the upper dermis, where melanoma cells may exist [52]. This results in better drug accumulation at the targeted site and enhances therapeutic effectiveness while reducing systemic exposure [42]. Furthermore, positively charged particles exhibit improved interactions with the negatively charged elements of the skin, increasing adhesion and retention at the application site—essential factors for treating aggressive skin cancers such as melanoma.

Moreover, the formulation’s acidic pH (~3.85) is within an acceptable range for topical application. The skin’s natural pH ranges from 4.5 to 5.5, and slightly acidic formulations can help maintain or restore the acid mantle, contributing to skin barrier function and antimicrobial defense. While prolonged exposure to low pH may cause irritation, short-term application of formulations around pH 3.8–4.0 has been reported to be well tolerated, especially in localized treatment settings such as melanoma therapy.

The drug content was 95 ± 1.8%, close to theoretical values (1 mg/mL) for nanocapsules containing PROP, indicating a high level of accuracy in the preparation process. The EE% was 81 ± 0.22%, which is considered high, especially considering that the drug used was in its hydrochloride salt form, known for its greater water solubility. Typically, the increased hydrophilicity of the hydrochloride form favors the drug’s partitioning into the aqueous phase during nanocapsule formation, which might reduce EE% in oil core-based systems. However, the high EE% obtained in this study suggests that the polymeric system used—particularly Eudragit RL^®^100—effectively retained the drug. This could be due to electrostatic interactions or the polymer’s ability to create microenvironments that accommodate both hydrophilic and lipophilic characteristics, stabilizing the propranolol hydrochloride within the polymeric matrix or at the particle–medium interface [43].

Furthermore, formulation parameters such as medium pH, the choice of oil, and solvent polarity may have limited the drug’s migration to the external aqueous phase, thereby enhancing its retention in the nanocapsules. This efficient entrapment is significant for controlled drug release applications, as it helps prevent initial burst release and ensures sustained delivery—a key advantage in the topical treatment of cutaneous melanoma, where prolonged local drug action is desired to improve therapeutic outcomes while minimizing systemic side effects.

### 3.2. In Vitro Drug Release

The in vitro release profile of PROP from the NC in acetate buffer, pH 5.5, was evaluated by comparison with the free PROP aqueous solutions (Figure 1). Free PROP demonstrates a rapid and nearly complete release within the first 3 h, achieving 100% release. In contrast, NC-PROP displays a sustained release profile, with a more gradual release in the initial hours and reaching close to 100% only after 8 to 12 h. This initial accelerated release can be attributed to the unencapsulated fraction of the drug, as evidenced by the EE% value of 81 ± 0.22%. The remaining encapsulated drug is progressively released as it diffuses through the polymeric matrix of the nanocapsules. Figure 1 shows that the release of both solutions starts from primary minutes, reaches the maximum content after 3 h, NC-PROP then reaches the plateau state at 8 h point, in a concentration of 98%, in comparison to free PROP that saturates in the environment. The difference between the NC-PROP form and free PROP solutions demonstrates that encapsulation of PROP prolongates the drug release time (*p* < 0.05), which enhances the drug effect’s duration, optimizing treatment with short half-life time drugs.

The sustained release profile observed for NC-PROP can be attributed to the polymeric barrier imposed by the Eudragit RL^®^100 wall, which limits the diffusion of the drug into the external environment and ensures prolonged delivery [53]. This behavior is advantageous in treating cutaneous melanoma, as the controlled release maintains therapeutic concentrations of the drug over an extended period at the application site, enhancing the antiproliferative effect on tumor cells and reducing the need for frequent reapplications, improving patient compliance compared to free PROP, which is rapidly released. Additionally, the slow release can minimize the systemic adverse effects associated with free propranolol, particularly those related to its systemic effects after cutaneous absorption, thereby reducing the risk of adverse cardiovascular effects such as bradycardia and hypotension [54]. Furthermore, the prolonged drug presence at the target site may improve tumor penetration and long-term pharmacodynamic activity, optimizing therapeutic outcomes. Therefore, the topical formulation with prolonged release may represent an effective and safe strategy for the local treatment of melanomas [55,56].

### 3.3. Antioxidant Activity

Oxidative stress plays a central role in carcinogenesis, promoting DNA damage, activating pro-inflammatory pathways, and impairing cellular defense mechanisms. Reactive oxygen species (ROS), generated as byproducts of normal oxygen metabolism, are usually neutralized by endogenous antioxidants. However, an imbalance that favors the accumulation of ROS can lead to oxidative stress, contributing to cellular dysfunction, malignant transformation, tumor progression, and resistance to therapies [57].

The ABTS radical scavenging assay is a widely used colorimetric method to evaluate antioxidant activity. In this test, the ABTS•⁺ radical cation generates a blue-green solution, and antioxidant compounds reduce the radical, causing discoloration. Thus, the higher the antioxidant activity, the greater the decrease in color intensity [58]. Figure 2 presents a bar chart that depicts the scavenging capacity (SC%) of propranolol (PROP), propranolol-loaded nanocapsules (NC-PROP), and blank nanocapsules (NC-B) at various concentrations (0.5 to 100 µg/mL). At lower concentrations (0.5–25 µg/mL), the SC% values were quite similar among the groups, with minor fluctuations (*p* > 0.05). However, at the highest concentration (50–100 µg/mL), NC-PROP showed a significantly greater antioxidant capacity (~105%) in comparison to free PROP (~52%) and NC-B (~90%) (*p* < 0.05).

In addition to its known β-blocking properties, propranolol has demonstrated antioxidant activity in previous studies, possibly related to its ability to inhibit lipid peroxidation and scavenge reactive species [59,60]. The aromatic structure of propranolol, containing hydroxylated groups, is believed to favor hydrogen donation or electron transfer, allowing the neutralization of free radicals such as ABTS•⁺ [61]. These findings support the hypothesis that propranolol may help reduce oxidative stress.

These findings suggest that the encapsulation of propranolol into nanocapsules enhances its antioxidant potential, especially at higher concentrations. This improvement may result from the synergistic effect between the antioxidant capacity of the drug and the components of the nanocapsule matrix, such as the polymer shell, oil, or surfactants, which can also scavenge reactive species [62]. Moreover, NC-PROP’s controlled release profile may prolong drug interaction with free radicals, increasing the cumulative antioxidant response.

This outcome is especially relevant for cutaneous applications in melanoma treatment. Oxidative stress is a critical factor in melanoma progression, contributing to DNA damage, tumor growth, and resistance to therapy. Therefore, topical formulations with improved antioxidant profiles may provide additional therapeutic benefits by mitigating oxidative damage in the tumor microenvironment [63]. Additionally, the enhanced SC% observed for NC-PROP supports its potential as a safer, more effective alternative to free propranolol for localized treatment, reducing the risk of systemic side effects while maintaining therapeutic efficacy [46,64].

### 3.4. Bioadhesiveness

The interaction of nanoparticles with mucin is widely used as an in vitro model to evaluate the bioadhesive potential of carrier systems. Although traditionally applied to simulate interactions with mucous membranes, the evaluation with mucin also provides relevant information for systems intended for cutaneous application. The skin, like mucous membranes, has a negatively charged surface, especially in the stratum corneum, favoring electrostatic interactions with cationic particles [65,66]. Thus, the ability to interact with mucin can be considered a preliminary indicator of the potential for cutaneous bioadhesion, a desirable characteristic to optimize topical retention and therapeutic efficacy of systems administered to the skin [30].

In the present study, the dispersion of NC-B and NC-PROP in 0.1% (*w*/*v*) mucin induced significant changes in their physicochemical properties, indicating a strong interaction between mucin and the nanocapsule surface (Figure 3). For NC-B, the particle size increased from 142.65 nm (in water) to 271.72 nm (in mucin), while for NC-PROP, it increased from 148.23 nm to 250.33 nm (*t*-test, *p* < 0.05). The polydispersity index (PDI) also increased significantly for both formulations, from values below 0.2 to above 0.5 (*t*-test, *p* < 0.05), indicating greater heterogeneity in the particle size distribution. Furthermore, a notable inversion in surface charge was observed: the zeta potential changed from +36.57 mV and +32.36 mV (in water) to −6.71 mV and −6.33 mV (in mucin) for NC-B and NC-PROP, respectively (*t*-test, *p* < 0.05). These findings suggest that negatively charged mucin molecules were adsorbed onto the positively charged surface of the nanocapsules, neutralizing or reversing the original charge. This electrostatic interaction likely resulted in the formation of a mucin coating, which is considered a key indicator of the bioadhesive potential of the formulations, enhancing their ability to interact with biological surfaces, such as negatively charged components of the skin barrier [30,67].

Bioadhesion is a critical attribute for formulations intended for cutaneous use, such as those with potential application in the treatment of melanoma, since it can increase residence time at the application site, improve drug penetration, and ensure sustained release. Prolonging the permanence of the nanocapsules on the skin can favor more stable therapeutic concentrations at the tumor site, reduce the frequency of application, and minimize systemic adverse effects [68,69,70]. Thus, the findings reinforce the applicability of the formulations developed for the topical treatment of melanoma, in line with the proposal to promote a more localized and effective treatment.

Finally, it is worth noting that the bioadhesive behavior observed is mainly due to the cationic nature of the Eudragit RL^®^100 polymer, used to form the polymeric wall of the nanocapsules [30]. Positively charged polymers are known to have a greater capacity to interact with negatively charged biological surfaces, conferring superior bioadhesive properties [29,71]. The presence of propranolol inside the nanocapsules (NC-PROP) did not compromise this interaction, indicating that bioadhesion is mainly governed by the characteristics of the coating polymer [53,67]. This behavior corroborates the choice of Eudragit RL^®^100 as a promising material for developing topical bioadhesive systems for treating melanoma.

### 3.5. Hemolysis Assay

Hemolysis assays are used to evaluate the biocompatibility and cytotoxicity of a nanoformulation [72,73,74]. Figure 4 illustrates the hemolytic effects of free PROP, NC-PROP, and NC-B. Generally, hemolysis below 5% is considered safe for blood cells [74]. Free PROP exhibits the lowest hemolytic activity, remaining under the 5% threshold across nearly all tested concentrations. Among the encapsulated forms, both presented a hemolytic effect above 50%. Despite this, NC-PROP demonstrates less hemolysis than NC-B.

The cytotoxic effect of the encapsulated forms is probably due to the Span^®^ 80 Tween^®^ 80 present in the composition, which can break the membrane of the cells. Furthermore, the marked rise in hemolysis observed in nanostructured formulations could be linked to the inclusion of cationic polymeric materials like Eudragit RL^®^100 [75,76]. These components may directly interact with the erythrocyte membrane, causing it to rupture. Additionally, the positive zeta potential of these formulations might further exacerbate this unfavorable interaction with the negatively charged surface of red blood cells [77].

Despite the widespread use of the hemolysis test, it is essential to consider that in vitro results may not directly reflect in vivo effects, especially for formulations intended for topical use. Studies indicate that the presence of plasma proteins and other blood components can form a “protein corona” around the nanoparticles, altering their interactions with cells and potentially reducing the in vitro hemolysis [76,78]. In addition, the topical route of administration greatly limits systemic exposure, minimizing the risk of direct contact with blood and, consequently, hemolysis. Even in cases of slight systemic absorption, dilution in the bloodstream would reduce potential adverse effects. Notably, although in vitro assays may indicate hemolytic potential, studies have shown that this does not necessarily translate into in vivo toxicity following topical application [79]. Thus, these findings should be interpreted in light of the intended cutaneous use.

### 3.6. Cell Viability Assay

The MTT reduction assay showed that all treatments were cytotoxic to both cell lines. For the L-929 cells, all the concentrations significantly reduced viability. When treatments are compared, NC-PROP was more cytotoxic than NC-B and PROP at higher concentrations (25 and 50 μg/mL) (Figure 5A). However, this effect was not observed at lower concentrations; NC-PROP shows even higher viability than NC-B. Regarding B16-F10 cells, only the higher concentrations were cytotoxic. NC-B significantly reduced cell viability at 50 μg/mL, while NC-PROP and PROP reduced cell viability at 25 and 50 μg/mL, respectively (Figure 5B). Similarly to the results for L-929, NC-PROP was more cytotoxic than other treatments at higher concentrations. The lower concentrations of all treatments caused an increase in cell viability.

Propranolol cytotoxicity has already been reported; its effects on hemangiomas are well documented, to the point that it is used in infantile hemangioma treatment protocols [80]. Other cancerous cell lines have been investigated as well, such as SCC-9, SCC-25 and Cal27 (oral squamous cell carcinoma) [81], A-549 (lung cancer) [82], MKN45 and NUGC3 (gastric cancer) [83], A375 (melanoma) [43]; and non-cancerous cells such as BEAS2B (lung) [84] and HUVECs (endothelial) [85]. Nanoformulations containing propranolol were also cytotoxic to several cell lines. A trehalosome loaded with propranolol was cytotoxic to A375 cells (melanoma), showing IC50 values lower than those of the drug and the blank formulation, which was also cytotoxic [43].

Some mechanisms were associated with cytotoxic activity, including the downregulation of p-P65 NF-κB and VEGF expression in SSC-9 and Cal27 cells, the inhibition of Akt and p70S6K phosphorylation, and an increase in PTEN phosphorylation in all OSCC cell lines [81]. In addition, apoptosis is induced through both caspase-dependent and -independent pathways in A459 cells [84]. In A375 melanoma cells, propranolol and its encapsulated nanoformulation increased both early and late apoptosis and induced G1 phase cell cycle arrest [43].

The cytotoxic effect observed for NC-B can be attributed to its formulation, which contains non-ionic surfactants, known for their dose-dependent cytotoxicity [86]. Since these compounds are amphiphilic, they can interact with cellular membranes and cause cell death by disrupting cell structures. The synergic effect of surfactants and drugs could explain the higher cytotoxic effect of NC-PROP. Similar results were observed for L-929 and B16-F10 cells treated with nanoparticles of a similar formulation; the non-loaded formulation was cytotoxic, although less intensely than the loaded ones [28].

On the other hand, the higher viability at lower concentrations could be due to cell proliferation or metabolic induction. Considering that the MTT assay is a colorimetric technique and depends on metabolic cell enzymes to reduce tetrazolium salt to formazan crystals, several compounds could interfere with it [87]. For example, lowering agents that reduce MTT in a non-enzymatic manner, and anticancer drugs that upregulate succinate dehydrogenase (SDH) and increase mitochondrial mass [88]. A scratch assay investigated whether lower concentrations have a proliferative or metabolic effect on B16-F10 cells.

### 3.7. Scratch Wound Assay

The scratch wound closure test can be used to elucidate different aspects of the treatment, such as healing potential and antiproliferative and antimetastatic activity [28,89,90]. In this study, we aimed to investigate the possible proliferative potential generated by the treatments, as lower concentrations showed increased cell viability. As measured by the MTT method, this increase may indicate either an increase in cell number or cell metabolism. When there is an increase in the number of cells, the tendency is for them to migrate and reduce the area of the groove initially formed [89]. However, none of the treatments in question generated an increase in proliferative activity at the lowest concentrations (0.5 and 1.0 µg/mL) since all of them presented a groove area similar to the negative control (*p* > 0.05), as demonstrated in Figure 6.

Therefore, this finding suggests that the increased viability observed in the MTT assay may have been due to increased cellular metabolism [91]. This report can be interpreted as a hormetic effect, characterized by increased cellular responses to low doses of toxic agents, followed by cytotoxicity at higher doses [92]. Previous research has shown that compounds in the propranolol pharmacological class can exhibit hormetic effects, which involve low-dose stimulation followed by inhibition at high doses. Understanding this biphasic nature is crucial for comprehending how these compounds interact with cellular processes, offering potential therapeutic advantages and risks depending on the concentration used. Propranolol has been observed to affect tumor growth in a dose-dependent manner. Studies indicate that propranolol has a biphasic effect, where lower doses might promote cellular survival and proliferation, while higher doses generally inhibit these processes [93]. Additionally, some anticancer medications show hormetic responses, with low doses fostering cell proliferation and high doses causing inhibition, highlighting the significance of dose in therapeutic approaches [92,94,95]. Moreover, the use of nanoformulations has gained interest regarding hormesis. Nanomaterials have been found to elicit hormetic responses in various biological systems, such as in studies involving plants and algae, where low concentrations can stimulate growth while higher levels lead to toxicity [96,97].

Propranolol has been shown to inhibit cell proliferation in different models, including gastric cancer cells (MKN45) at 200 µM and HEMEC endothelial cells at 50–75 µM, with greater effect at lower concentrations. These results, obtained under different culture conditions, suggest that nonselective blockade of β-adrenoreceptors may be related to the antitumor effect of PROP [45,83]. In a xenograft model of human melanoma, propranolol treatment significantly reduced tumor growth, cell proliferation, and intratumoral vascular density, thereby promoting tumor cell death [98]. However, although a reduction in viability was observed in the MTT assay, PROP was unable to prevent cell proliferation and migration at the tested concentrations.

In contrast, at higher concentrations (5 µg/mL), both NC-B and NC-PROP were effective in significantly inhibiting the migration of B16-F10 cells (Figure 7), maintaining approximately 80% of the initial groove area after 24 h (Figure 6, *p* < 0.05). Similar results were reported in studies with luteolin nanoparticles, which were shown to inhibit melanoma cell migration in groove closure assays [99,100].

Furthermore, unloaded formulations with a composition similar to NC-B have already shown an inhibitory potential on B16-F10 [28]. The ability of nanostructured formulations to inhibit cell migration is particularly relevant in the context of cutaneous melanoma, where preventing cell dissemination is crucial for controlling tumor progression [28,101]. Thus, our findings reinforce the therapeutic potential of the developed nanocapsules, not only for controlled drug release, but also for their direct action in inhibiting cell migration.

## 4. Conclusions

This study successfully formulated propranolol-loaded Eudragit RL^®^100 nanocapsules, achieving high encapsulation efficiency alongside favorable physicochemical properties, including an appropriate particle size (~151 nm) and a positive zeta potential, which indicate good colloidal stability. The formulations displayed controlled and extended release of PROP, improved antioxidant activity, and significant mucin bioadhesion, crucial for topical delivery. Although the hemolysis rate exceeded 50%, this issue is likely negligible in topical applications due to minimal systemic absorption. Cell viability and migration assays further demonstrated that the nanocapsules markedly decreased melanoma cell viability and hindered cell migration, highlighting their potential to prevent tumor progression. The findings strongly advocate for the therapeutic effectiveness of these nanoformulations in treating cutaneous melanoma. Future research on skin penetration, preclinical models, and clinical trials will be essential to validate the practical use of this treatment strategy, particularly in improving drug retention at the melanoma site while reducing off-target effects.

## Figures and Tables

**Figure 1 pharmaceutics-17-00718-f001:**
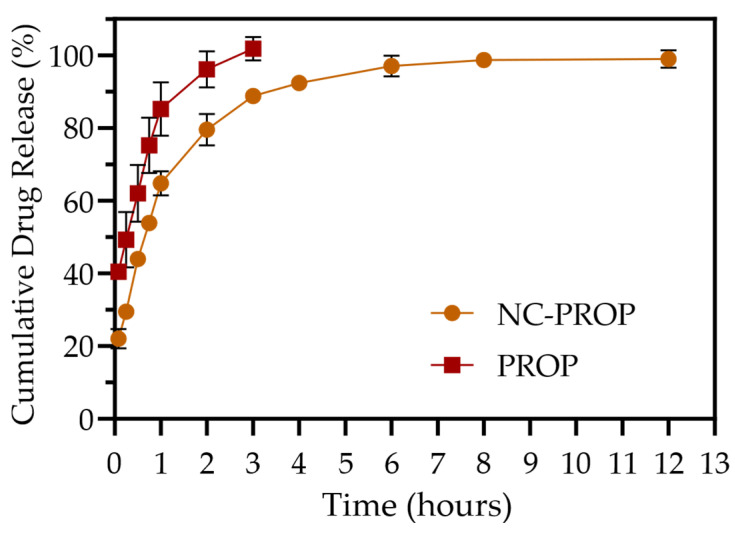
Cumulative in vitro drug release profiles of PROP and NC-PROP over 12 h in phosphate buffer (pH 5.5). Data are expressed as mean ± standard deviation (*n* = 3). NC-PROP and PROP differed within all concentrations (*p* < 0.05, two-way ANOVA, followed by Neuman–Keuls *post hoc* test).

**Figure 2 pharmaceutics-17-00718-f002:**
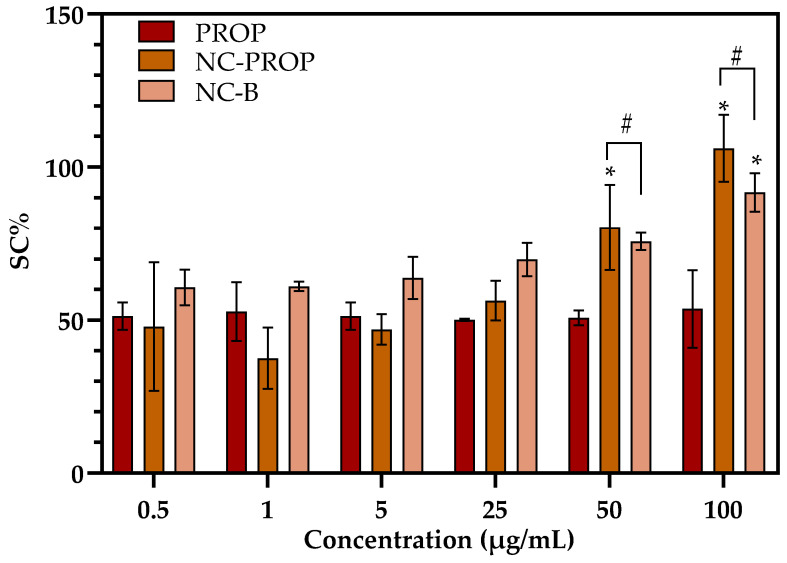
Scavenging capacity (SC%) of PROP, NC-PROP, and NC-B at increasing concentrations (0.5–100 µg/mL). Data are expressed as mean ± standard deviation (*n* = 3). Asterisks mean significant difference (*p* < 0.05) between same group in different concentrations, # stands for significant difference (*p* < 0.05) between different groups in same concentration by two-way ANOVA, followed by Neuman–Keuls *post hoc* test.

**Figure 3 pharmaceutics-17-00718-f003:**
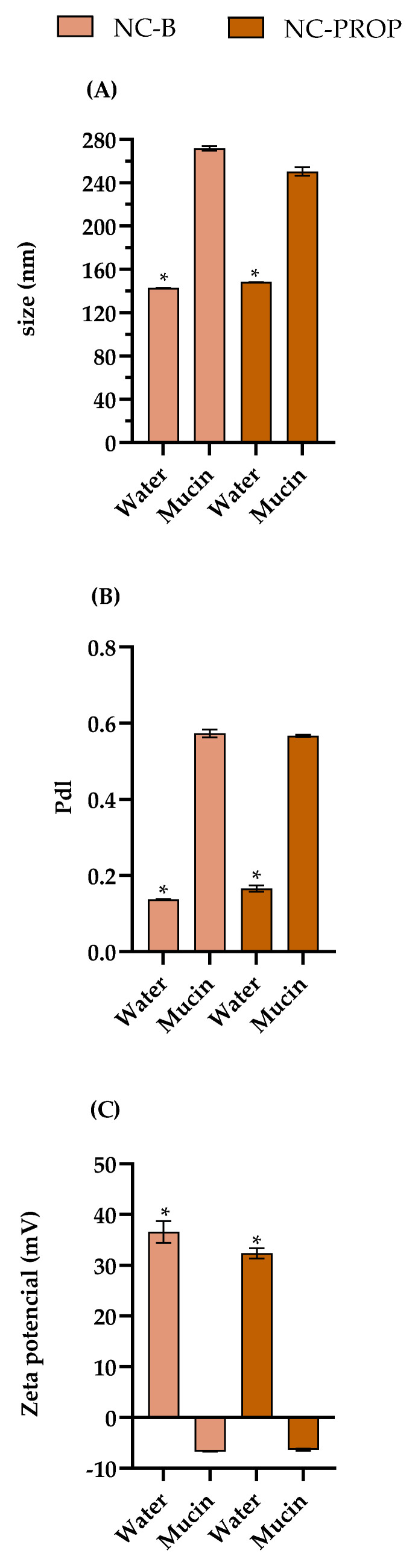
Effect of mucin (0.1%) on physicochemical characteristics of NC-B and NC-PROP. (**A**) Particle size, (**B**) polydispersity index (PDI), and (**C**) zeta potential. Asterisks mean significant difference before and after mucin interaction using *t*-test.

**Figure 4 pharmaceutics-17-00718-f004:**
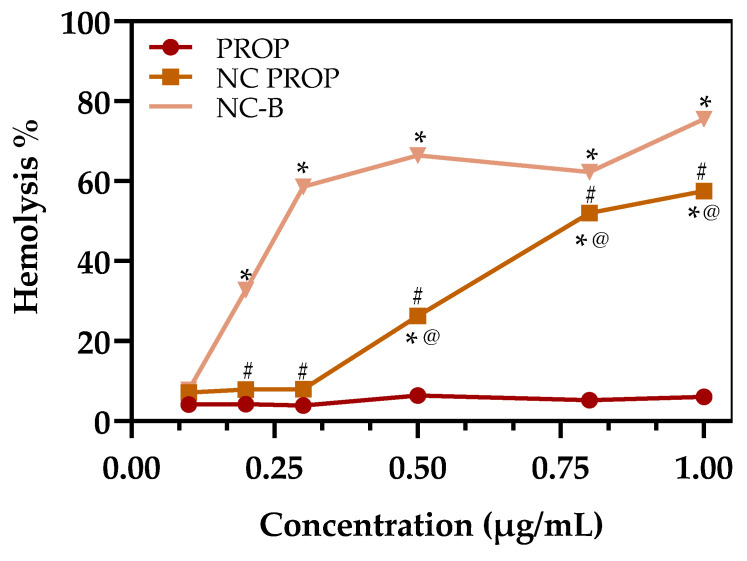
Hemolytic activity (%) of PROP, NC-B, and NC-PROP after incubation with human red blood cells. Results show low hemolysis for free PROP (<5%), while both NC-B and NC-PROP exhibited significantly increased hemolytic activity (>50%) at highest concentrations tested. Data are expressed as mean ± SD (*n* = 3). Asterisks mean significant difference between same group at different concentrations by two-way ANOVA, followed by Neuman–Keusl *post hoc* test. For NC-PROP, concentrations of 0.8 and 1 µg/mL were not different (*p* > 0.05). For NC-B, concentrations of 0.3–0.5 µg/mL and 0.5–0.8 µg/mL were not different (*p* > 0.05). # stands for significant difference between NC-PROP and NC-B (*p* < 0.05); @ represents significant difference between NC-PROP and PROP (*p* < 0.05).

**Figure 5 pharmaceutics-17-00718-f005:**
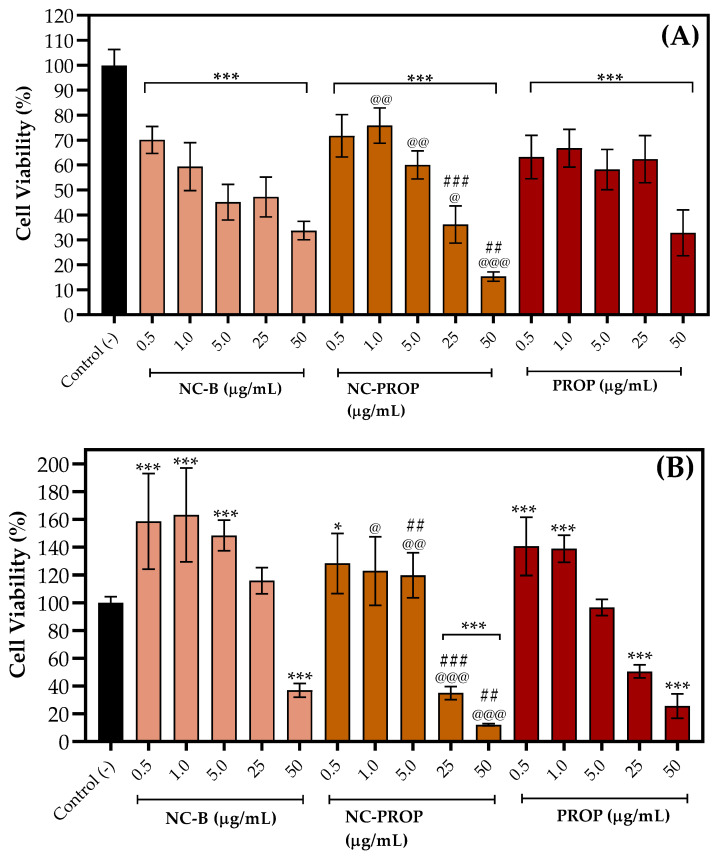
Cytotoxic effect of non-nanoencapsulated PROP, PROP-loaded nanocapsules (NC-PROP), and non-loaded nanocapsules (NC-B) against murine fibroblast cells (L929; (**A**)) and murine melanoma cells (B16-F10; (**B**)). Mean ± SD (*n* = 3). Asterisks denote significant differences (* *p* < 0.05 and *** *p* < 0.001) compared to negative control group as determined by one-way ANOVA followed by Newman–Keuls *post hoc* test. At symbol (@ *p* < 0.05, @@ *p* < 0.01, and @@@ *p* < 0.001) denotes significant differences between NC-B and NC-PROP in same concentration by unpaired *t*-test. Sharps (## *p* < 0.01 and ### *p* < 0.001) denote significant differences between PROP and NC-PROP in same concentration by unpaired *t*-test.

**Figure 6 pharmaceutics-17-00718-f006:**
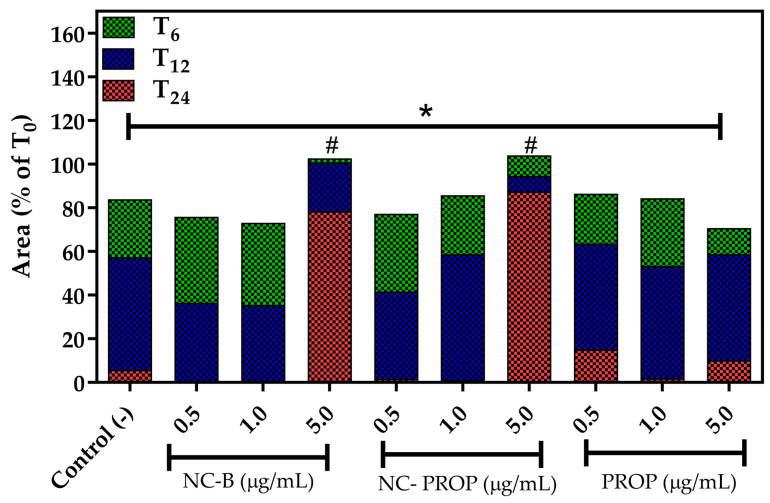
Wound area reduction in murine melanoma cells (B16-F10) after treatment with non-nanoencapsulated PROP, PROP-loaded nanocapsules (NC-PROP), and non-loaded nanocapsules (NC-B). Black bar denotes mean (*n* = 3). Asterisk indicates significant differences (*p* < 0.01) compared T_0_ and T_24_ as determined by one-way ANOVA followed by Newman–Keuls post hoc test. Sharp (# *p* < 0.01) denotes significant difference regarding negative control at same time of analysis determined by one-way ANOVA, followed by Newman–Keuls *post hoc* test.

**Figure 7 pharmaceutics-17-00718-f007:**
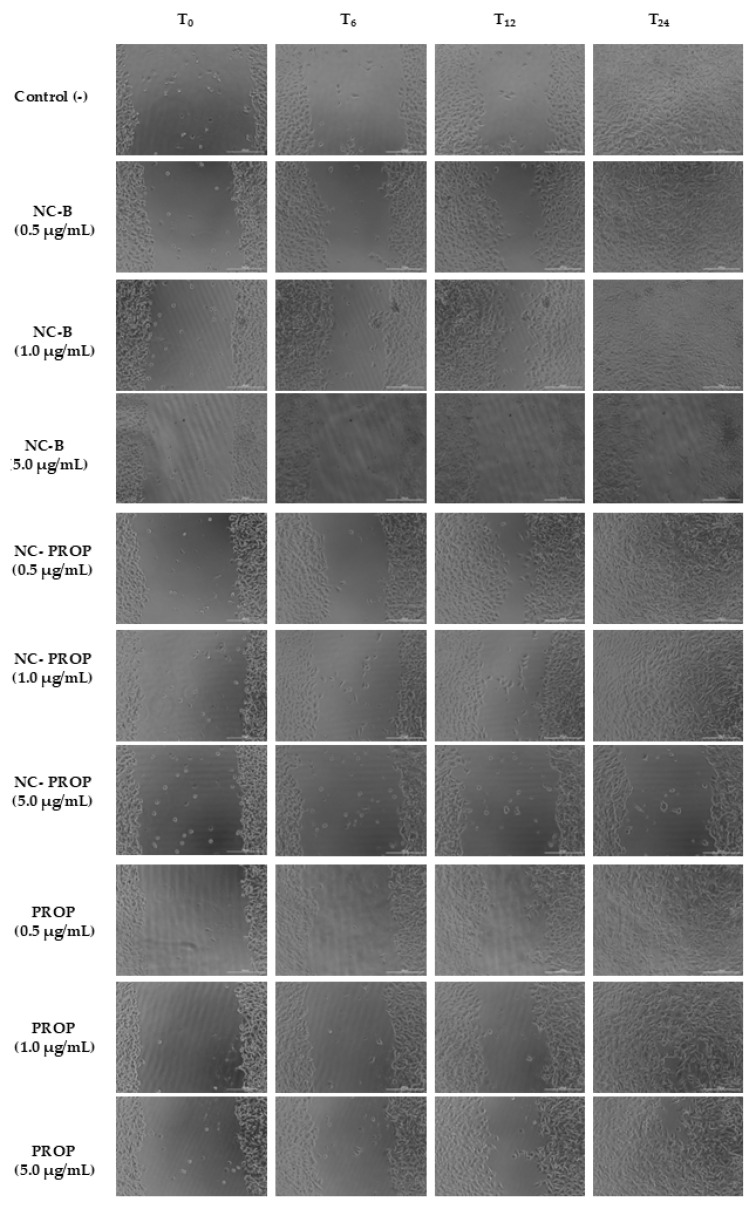
Wound area reduction on B16-F10 cells at 0, 6, 12, and 24 h after treatment with non-nanoencapsulated PROP, PROP-loaded nanocapsules (NC-PROP), and non-loaded nanocapsules (NC-B) (T: time). DMEM was used as control. Pictures were taken in phase contrast mode, 10×, scale bar = 200 μm.

**Table 1 pharmaceutics-17-00718-t001:** Physicochemical characterization.

Scheme	Particle Size	PDI	Zeta Potential	pH
NC-PROP	151 ± 7.86	0.134 ± 0.01	25.84 ± 4.89	3.85 ± 0.117
NC-B	150 ± 1.74	0.166 ± 0.03	34.13 ± 8.70	3.95 ± 0.181

## Data Availability

The original contributions presented in the study are included in the article; further inquiries can be directed to the corresponding author/s.

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
