# Peer review of "Bioadhesive Eudragit RL®100 Nanocapsules for Melanoma Therapy: A Repurposing Strategy for Propranolol"

_pharmaceutics, 2025, doi:10.3390/pharmaceutics17060718_

Round 1

Reviewer 1 Report

Comments and Suggestions for Authors

The manuscript describes loading of propranolol in eudragit nanoparticles for topical treatment of melanoma. The work would be of interest to the readers of pharmaceutics, and it addresses the knowledge gap of advancing the efficacy of repurposed drugs using nanotechnology. References are adequate in number and relevant to the described work. However, I have some questions/comments for the authors that need to be addressed:

1- The novelty of the work is not sufficiently highlighted in the introduction section. Please indicate if this is the first time that propranolol has been topically applied, and if that's the first study to be loaded in eudragit nanoparticles, and if that's the first study to repurpose it in melanoma treatment. 

2- Please state the pharmaceutical challenges for delivering propranolol in the introduction section.

3- Please replace the word "suspension" by "dispersion". Suspension by definition refers to micron-sized particles.

4- Table 1, was the difference in zeta potential between blank and drug loaded formulation significant? if yes an interpretation is needed.

5- Figure 2, 4, correct the X axis title

6- Correct the spelling of the word "mucine" to "mucin" in Figure 3

7- The IC50 values of the formulations need to reported as mean and S.D. and compared statistically. The authors need to justify why viability for cells exceeded 150% at some instances. Was there a microbial contamination causing further turbidity and giving erraneous results?

Author Response

# Reviewer 1

The manuscript describes loading of propranolol in eudragit nanoparticles for topical treatment of melanoma. The work would be of interest to the readers of pharmaceutics, and it addresses the knowledge gap of advancing the efficacy of repurposed drugs using nanotechnology. References are adequate in number and relevant to the described work. However, I have some questions/comments for the authors that need to be addressed:

Answer: We sincerely thank the reviewer for the encouraging feedback regarding the relevance and comprehensiveness of our manuscript. We appreciate the reviewer’s indication that further improvements are needed, and we have carefully addressed all specific comments and suggestions in the following responses. We believe these revisions have significantly strengthened the manuscript's quality, clarity, and scientific rigor.

1- The novelty of the work is not sufficiently highlighted in the introduction section. Please indicate if this is the first time that propranolol has been topically applied, and if that's the first study to be loaded in eudragit nanoparticles, and if that's the first study to repurpose it in melanoma treatment. 

Answer: We appreciate the reviewer's insightful comment. To address this issue, we revised the introduction section and made some modifications that are marked in red to clarify the novelty of the study (Lines 100-116).

2- Please state the pharmaceutical challenges for delivering propranolol in the introduction section.

Answer: We thank the reviewer for suggesting improvements in stating the pharmaceutical challenges for topical delivery of propranolol. We made some adjustments to clarify these challenges in lines 67-77.

3- Please replace the word "suspension" by "dispersion". Suspension by definition refers to micron-sized particles.

Answer: We sincerely appreciate the reviewer's suggestion and have replaced the word “suspension” by “dispersion” in every needed quote.

4- Table 1, was the difference in zeta potential between blank and drug loaded formulation significant? if yes an interpretation is needed.

Answer: We thank the reviewer for asking about the significant difference in Table 1. The t-test was applied, and no significance was shown. This information can be found on line 302.

5- Figure 2, 4, correct the X axis title

Answer: We thank the reviewer for this observation. We revised the figures and corrected the titles.

6- Correct the spelling of the word "mucine" to "mucin" in Figure 3

Answer: We sincerely appreciate the reviewer's observation; we revised and corrected the spelling.

7- The IC50 values of the formulations need to reported as mean and S.D. and compared statistically. The authors need to justify why viability for cells exceeded 150% at some instances. Was there a microbial contamination causing further turbidity and giving erraneous results?

Answer: We thank the reviewer for the suggestion and question presented. Classically, comparing treatments in cell cultures can be performed through IC₅₀ values or statistical analysis of the viability percentages observed at each concentration tested (Reolon et al., 2024; Ferreira et al., 2019). In the present study, we opted for the second approach and, therefore, did not calculate IC₅₀. This decision is due to the hormetic profile found: at the lowest concentrations, free propranolol increased viability to approximately 140%, while the nanoencapsulated formulation (NC-PROP) reached approximately 120%. As the IC₅₀ is defined with the response amplitude of each curve, differences in the “top” induced by metabolic stimulation would artificially shift the 50% point and introduce bias in the comparison between formulations.

Furthermore, the migration assay indicated no wound closure, suggesting that the increased signal reflects increased metabolic activity rather than effective cell proliferation. Thus, an IC₅₀ calculated on these data would primarily reflect the intensity of this metabolic stimulation rather than genuine differences in cytotoxicity. Therefore, we present the results as mean viability ± standard deviation for each concentration, comparing treatments by ANOVA followed by Tukey's test. This procedure provides a more transparent and robust assessment of the differences between formulations under identical experimental conditions.

Regarding the possibility of microbial contamination, we clarify that all activities strictly followed Good Cell Culture Practices. The formulations were prepared under aseptic conditions: glassware sterilized in an autoclave, use of ultrapure water, careful sanitization of benches and equipment, and complete equipment for the handler. All cell procedures were performed in a laboratory equipped with a laminar flow hood previously disinfected with 70% alcohol and subjected to a 30-min UVC light cycle. Plates, tips, and pipettes were sterile and disposable, minimizing additional risk (Pamies et al., 2022; ATCC, 2014; Reolon et al., 2024).

Before incubation with the MTT reagent, the plates were inspected macroscopically and microscopically to rule out contamination (ATCC, 2014). At the macroscopic level, no turbidity, lumps, or changes in color/pH of the medium were observed after 24 h of treatment. Under microscopy (Cytation 5, BioTek Instruments, Winooski, VT, USA), no wells showed bacteria, yeast, or fungi. These precautions, combined with repetition in three independent experiments, allow greater certainty in inferring that there was no contamination; therefore, the increase in viability/metabolism observed at low concentrations reflects an intrinsic effect of the formulations, a hypothesis reinforced by the results of the cell migration assay

References:

FERREIRA, L. M. et al. Xanthan gum-based hydrogel containing nanocapsules for cutaneous diphenyl diselenide delivery in melanoma therapy. Investigational New Drugs, 38: 662-674, 2020.

PAMIES, D. et al. Guidance document on Good Cell and Tissue Culture Practice 2.0 (GCCP 2.0). ALTEX, 39: 268-292, 2022.

REOLON, J. B. et al. Pomegranate oil-based nanocapsules enhance 3,3′-diindolylmethane action against melanoma cells. Brazilian Journal of Pharmaceutical Sciences, 60: e23931, 2024.

ATCC. Animal Cell Culture Guide. Manassas, VA: American Type Culture Collection, 2014.

Reviewer 2 Report

Comments and Suggestions for Authors

The manuscript focusing on the development of nanocapsule formulations of propranolol hydrochloride is interesting and matches the scope of the journal. A few sections of the article needs to be updated for additional information. The authors are suggested to address the below comments:

  1. What do authors mean by performed polymer (line 20)
  2. Maintain uniformity for font style “in vitro”
  3. Line 31-33: Authors claim that the nanocapsules without the drug (placebo) has also demonstrated therapeutic benefit. Can authors please cross check this statement.
  4. Section 2.2: what is the injection volume. Provide information for calibration curve, filter size used, LOD and LOQ.
  5. Section 2.3: was the API potency adjusted?
  6. Section 2.4: It would have been better if each methodology was divided into individual section
  7. Any justification for selecting the dialysis membrane with 10k molecular weight?
  8. Why was pH 5.5 buffer selected for release testing?
  9. Within the introduction please make it clear to the readers what you are trying to achieve by developing the nancapsule formulation.

Author Response

# Reviewer 2

The manuscript focusing on the development of nanocapsule formulations of propranolol hydrochloride is interesting and matches the scope of the journal. A few sections of the article needs to be updated for additional information. The authors are suggested to address the below comments:

Answer: We sincerely thank the reviewer for the encouraging feedback regarding the relevance and comprehensiveness of our manuscript. We appreciate the reviewer’s indication that further improvements are needed, and we have carefully addressed all specific comments and suggestions in the following responses. We believe these revisions have significantly strengthened the manuscript's quality, clarity, and scientific rigor.

  1. What do authors mean by performed polymer (line 20)

Answer: We thank the reviewer for this comment. However, we would like to clarify that the manuscript does not include the term “performed polymer.” The correct term used is “preformed polymer,” which refers to the fact that the polymer (Eudragit RL®100) was already synthesized and fully formed before its use in the formulation process. This terminology is commonly employed to indicate that the polymer was not synthesized in situ, but rather incorporated in its ready-to-use form during nanocapsule preparation.

  1. Maintain uniformity for font style “in vitro”

Answer:  We thank the reviewer for the suggestion to standardize the style of the “in vitro” term. We revised the work, and with the exception of the headings (mentioned in methodologies and results), all “in vitro” quotations have been standardized.

  1. Line 31-33: Authors claim that the nanocapsules without the drug (placebo) has also demonstrated therapeutic benefit. Can authors please cross check this statement.

Answer: We thank the reviewer for this important observation. We have carefully reviewed the statement and confirm that the nanocapsules without propranolol (NC-B) indeed exhibited biological activity in our in vitro assays. Specifically, in the cell viability and scratch wound assays, NC-B significantly reduced melanoma cell viability and inhibited cell migration at higher concentrations, as detailed in the Results section (Sections 3.7 and 3.8, Figures 5 and 6). These effects are likely related to the presence of surfactants and the cationic polymer Eudragit RL®100, both of which are known to interact with cellular membranes. While space limitations in the abstract prevented a more nuanced explanation, we have ensured that the full context and interpretation are provided in the body of the manuscript.

  1. Section 2.2: what is the injection volume. Provide information for calibration curve, filter size used, LOD and LOQ.

Answer: We sincerely thank the reviewers for pointing out the missing information in this section. The injection volume was 20µL, and we used a 0.45 µm filter. The calibration curve was constructed using concentrations ranging from 5 to 50µg/mL, and the LOD and LOQ were determined to be 1.130 and 3.425µg/mL, respectively. This information was added on lines 136-140.

  1. Section 2.3: was the API potency adjusted?

Answer: We thank the reviewer for this important question. In this study, the potency of propranolol hydrochloride was not adjusted. The active pharmaceutical ingredient (API) was used as received from the supplier (SM Empreendimentos, Brazil), following the specifications stated on the certificate of analysis and product label. The amount weighed for the formulation was based on the declared content, assuming a potency of 100%. This approach is commonly used in preformulation and early-stage development studies. However, we acknowledge that in future development stages, particularly for clinical or industrial translation, adjusting for API potency would be a necessary step to ensure dose accuracy.

  1. Section 2.4: It would have been better if each methodology was divided into individual section

Answer: We thank the reviewer for this constructive suggestion to improve the clarity and organization of the Methods section. In response, we have revised Section 2.4 by dividing the different methodologies into individual subsections, each with its own heading. We believe this new structure enhances the readability of the manuscript and allows readers to more easily locate and understand the specific experimental procedures.

  1. Any justification for selecting the dialysis membrane with 10k molecular weight?

Answer: We thank the reviewer for this pertinent question. The selection of the dialysis membrane with a 10 kDa molecular weight cutoff (MWCO) was based on its suitability to retain the polymeric nanocapsules while allowing the free propranolol molecules (molecular weight ≈ 259 Da) to diffuse through. This cutoff is widely used in nanoparticle release studies, as it effectively prevents nanoparticle leakage while enabling accurate monitoring of drug release kinetics. Additionally, this MWCO has been successfully employed in previous works involving nanocarriers and similar polymeric systems, supporting its appropriateness for our experimental design.

  1. Why was pH 5.5 buffer selected for release testing?

Answer: We thank the reviewer for this relevant question. The acetate buffer at pH 5.5 was selected to mimic the physiological pH of healthy human skin, particularly the stratum corneum, which typically ranges from pH 4.5 to 5.5. This choice ensures that the in vitro release profile more closely reflects the conditions under which the formulation is expected to be applied.

  1. Within the introduction please make it clear to the readers what you are trying to achieve by developing the nancapsule formulation.

Answer: We thank the reviewer for this valuable observation. In response, we have revised the final paragraph of the Introduction to more clearly highlight the study's main objective (Lines 107-116). The purpose of developing the propranolol-loaded nanocapsule formulation is to enable its topical use for the treatment of cutaneous melanoma by enhancing local drug retention, sustaining drug release at the tumor site, improving propranolol’s physicochemical stability, and minimizing systemic exposure and side effects. This strategy aims to repurpose propranolol as a safe and effective topical anticancer therapy, supported by the bioadhesive and controlled-release properties of the Eudragit RL®100 nanocapsules.